# ADAPTIVE DROPOUT WITH RADEMACHER COMPLEXITY REGULARIZATION

**Ke Zhai**
Microsoft AI & Research
Sunnyvale, CA
kezhai@microsoft.com

**Huan Wang**[*]
Salesforce Research
Palo Alto, CA
joyousprince@gmail.com

## ABSTRACT

We propose a novel framework to adaptively adjust the dropout rates for the deep neural network based on a Rademacher complexity bound. The state-of-the-art deep learning algorithms impose dropout strategy to prevent feature co-adaptation. However, choosing the dropout rates remains an art of heuristics or relies on empirical grid-search over some hyperparameter space. In this work, we show the network Rademacher complexity is bounded by a function related to the dropout rate vectors and the weight coefficient matrices. Subsequently, we impose this bound as a regularizer and provide a theoretical justified way to trade-off between model complexity and representation power. Therefore, the dropout rates and the empirical loss are unified into the same objective function, which is then optimized using the block coordinate descent algorithm. We discover that the adaptively adjusted dropout rates converge to some interesting distributions that reveal meaningful patterns. Experiments on the task of image and document classification also show our method achieves better performance compared to the state-of-the-art dropout algorithms.

## 1 INTRODUCTION

Dropout training (Srivastava et al., 2014) has been proposed to regularize deep neural networks for classification tasks. It has been shown to work well in reducing co-adaptation of neurons—and hence, preventing model overfitting. The idea of dropout is to stochastically set a neuron's output to zero according to Bernoulli random variables. It has been a crucial component in the winning solution to visual object recognition on ImageNet (Krizhevsky et al., 2012). Ever since, there have been many follow-ups on novel learning algorithms (Goodfellow et al., 2013; Baldi & Sadowski, 2013), regularization techniques (Wager et al., 2013), and fast approximations (Wang & Manning, 2013).

However, the classical dropout model has a few limitations. First, the model requires to specify the retain rates, i.e., the probabilities of keeping a neuron's output, a priori to model training. Subsequently, these retain rates are kept fixed throughout the training process thereafter. It is often not clear how to choose the retain rates in an optimal way. They are usually set via grid-search over hyper-parameter space or simply according to some rule-of-thumb. Another limitation is that all neurons in the same layer share the same retain rate. This exponentially reduces the search space of hyper-parameter optimization. For example, Srivastava et al. (2014) use a fixed retain probability throughout training for all dropout variables in each layer.

In this paper, we propose a novel regularizer based on the Rademacher complexity of a neural network (Shalev-Shwartz & Ben-David, 2014). Without loss of generality, we use multilayer perceptron with dropout as our example and prove its Rademacher complexity is bounded by a term related to the dropout probabilities. This enables us to explicitly incorporate the model complexity term as a regularizer into the objective function.

This Rademacher complexity bound regularizer provides us a lot of flexibility and advantage in modeling and optimization. First, it combines the model complexity and the loss function in an unified objective. This offers a viable way to trade-off the model complexity and representation

---

[*]Work done at Microsoft. Authors contribute equally.

power through the regularizer weighting coefficient. Second, since this bound is a function of dropout probabilities, we are able to incorporate them explictly into the computation graph of the optimization procedure. We can then adaptively optimize the objective and adjust the dropout probabilities throughout training in a way similar to ridge regression and the lasso (Hastie et al., 2009). Third, our proposed regularizer assumes a neuron-wise dropout manner and models different neurons to have different retain rates during the optimization. Our empirical results demonstrate interesting trend on the changes in histograms of dropout probabilities for both hidden and input layers. We also discover that the distribution over retain rates upon model convergence reveals meaningful pattern on the input features.

To the best of our knowledge, this is the first ever effort of using the Rademacher complexity bound to adaptively adjust the dropout probabilities for the neural networks. We organize the rest of the paper as following. Section 2 reviews some past approaches well aligned with our motivation, and highlight some major difference to our proposed approach. We subsequently detail our proposed approach in Section 3. In Section 4, we present our thorough empirical evaluations on the task of image and document classification on several benchmark datasets. Finally, Section 5 concludes this paper and summarizes some possible future research ideas.

## 2  RELATED WORKS

There are several prior works well aligned with our motivation and addressing similar problems, but significantly different from our method. For example, the standout network (Ba & Frey, 2013) extends dropout network into a complex network structure, by interleaving a binary belief network with a regular deep neural network. The binary belief network controls the dropout rate for each neuron, backward propagates classification error and adaptively adjust according to training data. Zhuo et al. (2015) realize the dropout training via the concept of Bayesian feature noising during neural network learning. They further extend the model to incorporate either dimension-specific or group-specific noise and propose framework to adaptively learn the dropout rates. Li et al. (2016) sample dropout from a multinomial distribution on neuron basis and establish a risk bound for stochastic optimization. They then propose the evolutional dropout model to adaptively update the sampling probabilities during training time.

In addition to these approaches, one other family of solution is via the concept of regularizer. Wang & Manning (2013) propose fast approximation methods to marginalize the dropout layer and show that the classical dropout can be approximated by a Gaussian distribution. Later, Wager et al. (2013) show that the dropout training on generalized linear models can be viewed as a form of adaptive regularization technique. Gal & Ghahramani (2016) develop a new theoretical framework casting dropout training as approximation to Bayesian inference in deep Gaussian processes. It also provides a theoretical justification and formulates dropout into a special case of Bayesian regularization. In the mean time, Maeda (2014) discusses a Bayesian perspective on dropout focusing on the binary variant, and also demonstrate encourage experimental results. Generalized dropout (Srinivas & Babu, 2016) further unifies the dropout model into a rich family of regularizers and propose a Bayesian approach to update dropout rates.

One popular method along with these works is the variational dropout method (Kingma et al., 2015), which provides an elegant interpretation of Gaussian dropout as a special case of Bayesian regularization. It also proposes a Bayesian inference method using a local reparameterization technique and translates uncertainty of global parameters into local noise. Hence, it allows inference on the parameterized Bayesian posteriors for dropout rates. This allows us to adaptively tune individual dropout rates on layer, neuron or even weight level in a Bayesian manner. Recently, Molchanov et al. (2017) extend the variational dropout method with a tighter approximation which subsequently produce more sparse dropout rates. However, these models are fundamentally different than our proposed approach. They directly operates on the Gaussian approximation of dropout models rather than the canonical multiplicative dropout model, whereas our proposed method directly bounds the model complexity of classical dropout model.

Meanwhile, the model complexity and the generalization capability of deep neural networks have been well studied in theoretical perspective. Wan et al. (2013) prove the generalization bound for the DropConnect neural networks—a weight-wise variant of dropout model. Later, Gao & Zhou (2016) extend the work and derive a Rademacher complexity bound for deep neural networks with dropout.

Another perspective to the model generalization is the PAC-Bayes bound proposed by McAllester (2013). The PAC-Bayes method assumes probability measures on the hypothesis space, and gives generalization guarantee over all possible "priors". McAllester (2013) give a PAC-Bayes bound for linear predictors with dropout. In practise, the PAC-Bayes method has the potential to give a even tigher generalization bound. The bound we prove in this paper is based on traditional techniques using Rademacher complexity. It is a first step towards understanding how the dropout method works, and we would like to extend it to the PAC-Bayes paradigm in the future.

These works provide a theoretical guarantee and mathematical justification on the effectiveness of dropout method in general. However, they all assume that all input and hidden layers have the same dropout rates. Thus their bound can not be applied to our algorithm.

## 3 RADEMACHER COMPLEXITY REGULARIZATION

We would like to focus on the classification problem and use multilayer perceptron as our example. However, note that the similar idea could be easily extended to general feedforward networks. Let us assume a labeled dataset $\mathbb{S} = \{(\mathbf{x}_i, \mathbf{y}_i) | i \in \{1, 2, \ldots, n\}, \mathbf{x}_i \in \mathbb{R}^d, \mathbf{y_i} \in \{0, 1\}^k\}$, where $\mathbf{x}_i$ is the feature of the $i^{\text{th}}$ sample, $\mathbf{y_i}$ is the one-hot class label for the $i^{\text{th}}$ sample, and $k$ is the number of classes in prediction. Without loss of generality, an $L$-layer multilayer perceptron with dropout can be modeled as a series of recursive function compositions. Let $k^l$ be the number of neurons of the $l^{\text{th}}$ layer. In particular, the first layer takes sample features as input, i.e., $k^0 = d$, and the last layer outputs the prediction, i.e., $k^L = k$.

We denote $\mathbf{W}^l \in \mathbb{R}^{k^{l-1} \times k^l}$ as the linear coefficient matrix from the $(l-1)^{\text{th}}$ layer to the $l^{\text{th}}$ layer, and $\mathbf{W}_i^l$ be the $i^{\text{th}}$ column of $\mathbf{W}^l$. For dropout, we denote $\boldsymbol{\theta}^l \in [0, 1]^{k^l}$ as the vector of retain rates for the $l^{\text{th}}$ layer. We also define $\mathbf{r}^l \in \{0, 1\}^{k^l}$ as a binary vector formed by concatenating $k^l$ independent Bernoulli dropout random variables, i.e., $r_j^l \sim \text{Bernoulli}(\theta_j^l)$. To simplify our notation, we further refer $\mathbf{W}^{:l} = \{\mathbf{W}^1, \ldots, \mathbf{W}^l\}$, $\mathbf{r}^{:l} = \{\mathbf{r}^0, \ldots, \mathbf{r}^l\}$, $\boldsymbol{\theta}^{:l} = \{\boldsymbol{\theta}^0, \ldots, \boldsymbol{\theta}^l\}$, $\mathbf{W} = \mathbf{W}^{:L}$, $\mathbf{r} = \mathbf{r}^{:(L-1)}$, and $\boldsymbol{\theta} = \boldsymbol{\theta}^{:(L-1)}$.

For an input sample feature vector $\mathbf{x} \in \mathbb{R}^d$, the function before the activation of the $j^{th}$ neuron in the $l^{th}$ layer $f_j^l$ is

$$f_j^l(\mathbf{x}; \mathbf{W}^{:l}, \mathbf{r}^{:l}) = \sum_t W_{tj}^l r_t^{l-1} \phi(f_t^{l-1}(\mathbf{x}; \mathbf{W}^{:l-1}, \mathbf{r}^{:l-1})), \forall l \in \{2, 3, \ldots, L\}$$

where $\phi : \mathbb{R} \to \mathbb{R}^+$ is the rectified linear activation function (Nair & Hinton, 2010, *ReLU*). In vector form, if we denote $\odot$ as the Hadamard product, we could write the output of the $l^{\text{th}}$ layer as

$$f^l(\mathbf{x}; \mathbf{W}, \mathbf{r}) = \left(\mathbf{r}^{l-1} \odot \phi(f^{l-1}(\mathbf{x}; \mathbf{W}^{:l-1}, \mathbf{r}^{:l-1}))\right) \mathbf{W}^l.$$

Without loss of generality, we also apply Bernoulli dropout to the input layer parameter $\boldsymbol{\theta}^0 \in \mathbb{R}^d$, i.e., $\mathbf{f}^1(\mathbf{x}; \mathbf{W}, \mathbf{r}^0) = (\mathbf{r}^0 \odot \mathbf{x}) \mathbf{W}^1$. Note that the output of the neural network $f^L(\mathbf{x}; \mathbf{W}, \mathbf{r}) \in \mathbb{R}^k$ is a random vector due to the Bernoulli random variables $\mathbf{r}$. We use the expected value of $f^L(\mathbf{x}; \mathbf{W}, \mathbf{r})$ as the deterministic output

$$f^L(\mathbf{x}; \mathbf{W}, \boldsymbol{\theta}) = \mathbb{E}_r[f^L(\mathbf{x}; \mathbf{W}, \mathbf{r})]. \tag{1}$$

The final predictions are made through a softmax function, and we use the cross-entropy loss as our optimization objective. To simplify our analysis, we follow Wan et al. (2013) and reformulate the cross-entropy loss on top of the softmax into a single logistic function

$$\text{loss}(f^L(\mathbf{x}; \mathbf{W}, \boldsymbol{\theta}), \mathbf{y}) = -\sum_j y_j \log \frac{e^{f_j^L(\mathbf{x}; \mathbf{W}, \boldsymbol{\theta})}}{\sum_j e^{f_j^L(\mathbf{x}; \mathbf{W}, \boldsymbol{\theta})}}.$$

### 3.1 EMPIRICAL RADEMACHER COMPLEXITY

**Definition** The empirical Rademacher complexity of function class $\mathbb{F}$ with respect to the sample $\mathbb{S}$ is

$$R_{\mathbb{S}}(\mathbb{F}) = \frac{1}{n} \mathbb{E}_{\{\sigma_i\}} \left[ \sup_{f \in \mathbb{F}} \sum_{i=1}^n \sigma_i f(s_i)) \right]$$

Define $\mathsf{loss} \circ f^L$ as the composition of the logistic loss function $\mathsf{loss}$ and the neural function $f^L$ returned from the $L^{th}$ (last) layer, i.e.,

$$\mathsf{loss} \circ f^L = \{(\mathbf{x}, \mathbf{y}) \rightarrow \mathsf{loss}(f^L(\mathbf{x}; \mathbf{W}, \boldsymbol{\theta}), \mathbf{y})\}.$$

**Theorem 3.1.** *Let* $\mathbf{X} \in \mathbb{R}^{n \times d}$ *be the sample matrix with the* $i^{th}$ *row* $\mathbf{x}_i \in \mathbb{R}^d$, $p \geq 1$, $\frac{1}{p} + \frac{1}{q} = 1$. *If the* $p$-*norm of every column of* $\mathbf{W}^l$ *is bounded by a constant* $B^l$, *denote* $\mathbb{W} = \{\mathbf{W} \mid \max_j \|\mathbf{W}_j^l\|_p \leq B^l, \forall l \in \{1, 2, \ldots, L\}\}$, *given* $\boldsymbol{\theta}$, *the empirical Rademacher complexity of the loss for the dropout neural network defined above is bounded by*

$$R_{\mathbb{S}}(\mathsf{loss} \circ f^L) = \frac{1}{n}\mathbb{E}_{\{\sigma_i\}} \left[\sup_{\mathbf{W} \in \mathbb{W}} \sum_{i=1}^n \sigma_i \mathsf{loss}(f^L(\mathbf{x}_i; \mathbf{W}, \boldsymbol{\theta}), \mathbf{y}_i)\right]$$
$$\leq k2^L \sqrt{\frac{2\log(2d)}{n}} \|\mathbf{X}\|_{max} \left(\Pi_{l=1}^L B^l \|\boldsymbol{\theta}^{l-1}\|_1^{1/q}\right),$$

*where* $k$ *is the number of classes to predict,* $\boldsymbol{\theta}^l$ *is the* $k^l$-*dimensional vector of Bernoulli parameters for the dropout random variables in the* $l^{th}$ *layer,* $\sigma_i s$ *are i.i.d. Rademacher random variables, and* $\|\cdot\|_{max}$ *is the matrix max norm defined as* $\|\mathbf{A}\|_{max} = \max_{ij} |A_{ij}|$.

Please refer to the appendix for the proof.

Theorem 3.1 suggests that the empirical Rademacher complexity of the dropout network specified in this paper is related to several terms in a multiplicative way:

i: $p$-norms of the coefficients: $\max_j \|\mathbf{W}_j^l\|_p$. Note that in (Srivastava et al., 2014), 2 norms of the coefficients are already used as regularizers in the experimental comparison

ii: 1-norms of the retain rates $\theta^l$

iii: sample related metrics: dimension of the sample $d$, the number of samples $n$, and maximum entries in the samples $X$

iv: the number of classes in the prediction $k$

An the extreme case is, if the retain rates $\theta^l$ for one layer are all zeros, then the upper bound above is tight, since in this case the network is simply doing random guess for predictions. Similarly when the coefficients in one layer are all zeros, the bound is also tight. In both cases the features from the samples are not even used in the prediction due to either zero retain rates or zero coefficients.

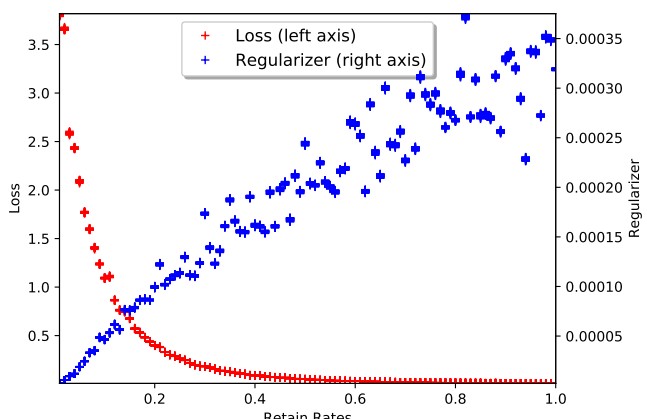

Figure 1: Empirical cross-entropy loss (left) and Rademacher regularizer (right) as a function of retain rates. We observe that the empirical loss and Rademacher regularizer change roughly in a monotonic way as a function of retain rates on training data. The experiments are evaluated on MNIST dataset with a hidden layer of 128 *ReLU* units. We apply dropout on the hidden layer only, and keep the retain rates fixed throughout training. We optimize with the empirical loss $\mathsf{Loss}(\mathbb{S}, f^L(\cdot; \mathbf{W}, \boldsymbol{\theta}))$, i.e., without any regularizer for 200 epochs with minibatch fo 100. All Rademacher regularizers are computed after every epoch in post-hoc manner, under the settings of $p = \infty, q = 1$. We plot the samples from last 20 epochs under each settings, with initial learning rate of 0.01, and decay by half every 40 epochs.

## 3.2 REGULARIZE WITH RADEMACHER COMPLEXITY

We have shown that the Rademacher complexity of a neural network is bounded by a function of the dropout rates, i.e., Bernoulli parameters $\boldsymbol{\theta}$. This makes it possible to unify the dropout rates and the network coefficients $\mathbf{W}$ in one objective. By imposing our upper bound of Rademacher complexity to the loss function as a regularizer, we have

$$\mathsf{Obj}(\mathbf{W}, \boldsymbol{\theta}) = \mathsf{Loss}(\mathbb{S}, f^L(\cdot; \mathbf{W}, \boldsymbol{\theta})) + \lambda \mathsf{Reg}(\mathbb{S}, \mathbf{W}, \boldsymbol{\theta}) \quad (2)$$

where the variable $\lambda \in \mathbb{R}^+$ is a weighting coefficient to trade off the training loss and the generalization capability. The empirical loss $\mathsf{Loss}(\mathbb{S}, f^L(\cdot; \mathbf{W}, \boldsymbol{\theta}))$ and regularizer function $\mathsf{Reg}(\mathbb{S}, \mathbf{W}, \boldsymbol{\theta})$ are defined as

$$\mathsf{Loss}(\mathbb{S}, f^L(\cdot; \mathbf{W}, \boldsymbol{\theta})) = \tfrac{1}{n} \sum_{(\mathbf{x}_i, \mathbf{y}_i) \in \mathbb{S}} \mathsf{loss}(f^L(\mathbf{x}_i; \mathbf{W}, \boldsymbol{\theta}), \mathbf{y}_i),$$

$$\mathsf{Reg}(\mathbb{S}, \mathbf{W}, \boldsymbol{\theta}) = k2^L \sqrt{\tfrac{\log d}{n}} \|\mathbf{X}\|_{max} \left( \Pi_{l=1}^L \|\boldsymbol{\theta}^{l-1}\|_1^{1/q} \max_j \|\mathbf{W}_j^l\|_p \right),$$

where $\mathbf{W}_j^l$ is the $j^{\text{th}}$ column of $\mathbf{W}^l$ and $\boldsymbol{\theta}^l$ is the retain rate vector for the $l^{\text{th}}$ layer. The variable $k$ is the number of classes to predict and $\mathbf{X} \in \mathbb{R}^{n \times d}$ is the sample matrix.

In addition to the Rademacher regularizer $\mathsf{Reg}(\mathbb{S}, \mathbf{W}, \boldsymbol{\theta})$, the empirical loss term $\mathsf{Loss}(\mathbb{S}, f^L(\cdot; \mathbf{W}, \boldsymbol{\theta}))$ also depends on the dropout Bernoulli parameters $\boldsymbol{\theta}$. Intuitively, when $\boldsymbol{\theta}$ becomes smaller, the loss term $\mathsf{Loss}(\mathbb{S}, f^L(\cdot; \mathbf{W}, \boldsymbol{\theta}))$ becomes larger, since the model is less capable to fit the training samples (i.e., less representation power), the empirical Rademacher complexity bound becomes smaller (i.e., more generalizable), and vice versa. Figure 1 plots the cross-entropy loss and empirical Rademacher $p = \infty, q = 1$ regularizer upon model convergence under different settings of retain rates. In the extreme case, when all $\boldsymbol{\theta}_j^l$ become zeros, the model always makes random guess for prediction, leading to a large fitness error $\mathsf{Loss}(\mathbb{S}, f^L(\cdot; \mathbf{W}, \boldsymbol{\theta}))$, and the Rademacher complexity $R_{\mathbb{S}}(\mathsf{loss} \circ f^L)$ approaches $0$.[1]

### 3.3 Optimize Dropout Rates

We now incorporate the Bernoulli parameters $\boldsymbol{\theta}$ into the optimization objective as in Eqn. (2), i.e., the objective is a function of both weight coefficient matrices $\mathbf{W}$ and retain rate vectors $\boldsymbol{\theta}$. In particular, the model parameters and the dropout rates are optimized using a block coordinate descent algorithm. We start with an initial setting of $\mathbf{W}$ and $\boldsymbol{\theta}$, and optimize $\mathbf{W}$ and $\boldsymbol{\theta}$ in an alternating fashion.

For the retain rate probability $\boldsymbol{\theta}$, due to the stochastic nature of dropout framework, it is very expensive to compute the exact objective value. For each training instance, we may have to exhaustively enumerate all possible dropout configurations in a combinatorial search space and compute its expectation of all the objective functions. One possbile approximation is to iteratively taking large number of samples from Bernoulli distributions of all layers for any input data and then compute the average objective. Even though, the computational complexity can be exponential as to the number of training data. Therefore, in our case, during the optimization of $\boldsymbol{\theta}$, we use the expected value of the Bernoulli dropout variables to rescale the output from each layer, to approximate the true objective $f^L(\mathbf{x}; \mathbf{W}, \boldsymbol{\theta})$. It significantly speeds up the forward propagation

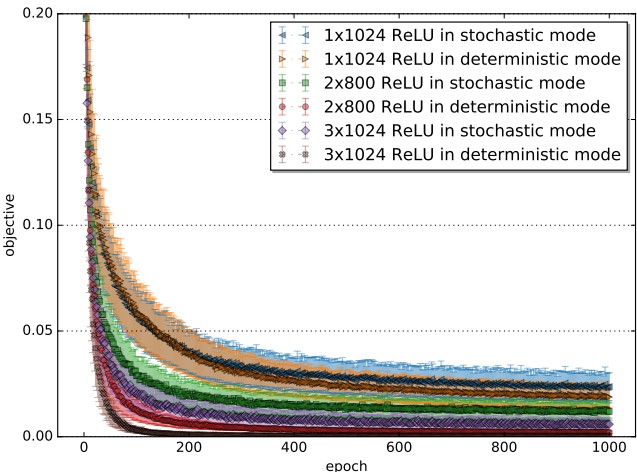

Figure 2: Changes in the true objectives in "stochastic" mode and their "deterministic" approximations against training epochs under different network architectures. The optimization objectives are reported on the training set of MNIST dataset, with Rademacher regularizer. We use minibatch size of $100$, initial learning rate of $0.01$, and decay it by half every $200$ epochs. The network structures we evaluated includes $1$ hidden layer with $1024$ units, $2$ hidden layers with $800$ units each, and $3$ hidden layers with $1024$ units each. The regularizer weights are set to $1e^{-3}$, $1e^{-4}$ and $1e^{-5}$ respectively. All neurons are *ReLU* units. Empirically, we find that optimizing the "deterministic" objective leads to similar improvements on the true "stochastic" objective as in Eqn. (2).

---

[1]To balance out the regularizer and the loss function so that they scale similarly as the sample size and internal layer nodes grow, we add some scaling factors to $\lambda$, which is discussed in Section 4.

process, as we do not need to iteratively sample the dropout variables for each training example. Essentially, it makes the layer output deterministic and the underlying network operates as if without dropout, which is exactly the same as the approximation used in (Srivastava et al., 2014) during testing time.

Note that using the expected value of Bernoulli dropout random variables to rescale a layer output is an approximation to the true objective $f^L(\mathbf{x}; \mathbf{W}, \boldsymbol{\theta})$. In practice, we find such "deterministic" approximation exhibits similar behavior during model optimization, and hence does not deviate or alter the performance, but significantly improves the running time. Figure 2 shows the true objective in "stochastic" mode and its "deterministic" approximation on the training set during optimization process under different network architectures. Empirically, we observe that the true optimization objective in stochastic mode as in Eqn. (2) decreases consistently if we use the expected value of the Bernoulli dropout random variable to approximate the sampling process.

# 4 EXPERIMENTS

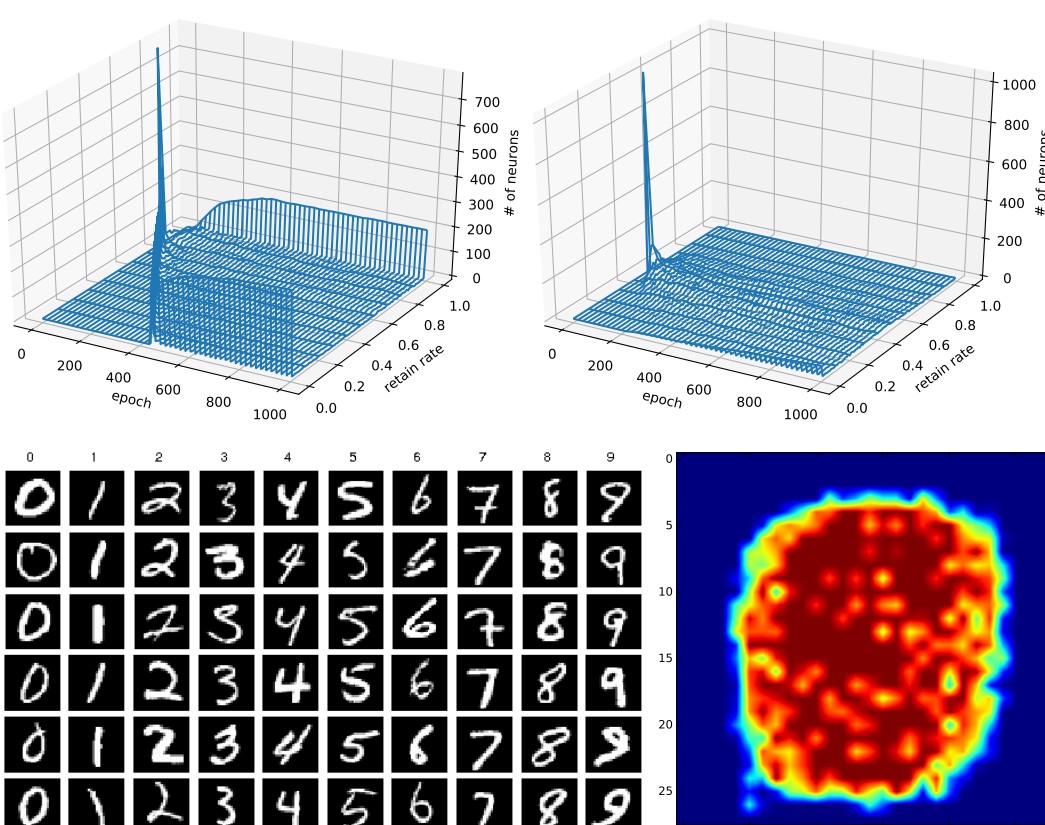

Figure 3: Changes in retain rates with Rademacher regularization on MNIST dataset. Top-Left: changes in retain rate histograms for input layer ($784$ gray scale pixels) through training. The retain rates get diffused over time, and only a handful of pixels have retain rates close to $1$. Top-Right: changes in retain rate histograms for hidden layer ($1024$ *ReLU* units) through training process. Bottom-Left: sample images from MNIST dataset. Bottom-Right: retain rates for corresponding input pixels upon model convergence. The surrounding pixels of input image yield smaller retain rates (corresponds to the dark background area), and the center ones have significantly larger retain rates (corresponds to the number pixels).

We apply our proposed approach with different network architectures, on the task of image and text classification using several public available benchmark datasets. All hidden neurons and convolutional filters are rectified linear units (Nair & Hinton, 2010, *ReLU*). We found that our approach achieves superior performance against strong baselines on all datasets. For all datasets, we hold out $20\%$ of the training data as validation set for parameter tuning and model selection. After then, we combine

both of these two sets to train the model and report the classification error rate on test set. We optimize categorical cross-entropy loss on predicted class labels with Rademacher regularization. In the context of this paper, we specifically refer the Rademacher regularizer to the case of $p = \infty, q = 1$ unless stated otherwise. We update the parameters using mini-batch stochastic gradient descent with Nesterov momentum of $0.95$ (Sutskever et al., 2013).

For Rademacher complexity term, we perform a grid search on the regularization weight $\lambda \in \{0.05, 0.01, 0.005, 0.001, 1e^{-4}, 1e^{-5}\}$, and update the dropout rates after every $I \in \{1, 5, 10, 50, 100\}$ minibatches. For variational dropout method (Kingma et al., 2015, VARDROP), we examine the both Type-A and Type-B variational dropout with per-layer, per-neuron or per-weight adaptive dropout rate. We found the neuron-wise adaptive regularization on Type-A variational dropout layer often reports the best performance under most cases. We also perform a grid search on the regularization noise parameter in $\{0.1, 0.01, 0.001, 1e^{-4}, 1e^{-5}, 1e^{-6}\}$. For sparse variational dropout method (Molchanov et al., 2017, SPARSEVARDROP), we find the model is much more sensitive to regularization weights, and often gets diverged. We examine different regularization weight in $\{1e^{-3}, 1e^{-4}, 1e^{-5}\}$. We follow similar weight adjustment scheme and scale it up by 10 after first $\{100, 200, 300\}$ epochs, then further scale up by 5 and 2 after same number of epoch.

**Scales of Regularization**    In practice, we want to stablize regularization term within some managable variance, so its value does not vary significantly upon difference structure of the underlying neural networks. Hence, we design some heuristics to scale the regularizer to offset the multipler effects raised from network structure. For instance, recall the neural network defined in Section 3, the Rademacher complexity regularizer with $p = \infty, q = 1$ after scaling is

$$k2^L \sqrt{\frac{\log d}{n}} \max_i \|x_i\|_\infty \left( \Pi_{l=1}^L \|\mathbf{W}^l\|_{max} \|\boldsymbol{\theta}^{l-1}\|_1 \frac{\sqrt{k^l + k^{l-1}}}{k^l} \right). \tag{3}$$

where $\mathbf{W}_j^l$ is the $j^{\text{th}}$ column of the weight coefficient matrix $\mathbf{W}^l$ and $\boldsymbol{\theta}^l$ is the retain rate vector for the $l^{\text{th}}$ layer. The variable $k$ is the number of classes to predict and $\mathbf{X} \in \mathbb{R}^{n \times d}$ is the sample matrix. Similarly, we could rescale the Rademacher complexity regularizers under other settings of $p = 2, q = 2$. Please refer to the appendix for the scaled Rademacher complexity bound regularizers and detailed derivations.

## 4.1 MNIST

MNIST dataset is a collection of $28 \times 28$ pixel hand-written digit images in grayscale, containing $60K$ for training and $10K$ for testing. The task is to classify the images into 10 digit classes from 0 to 9. All images are flattened into 784 dimension vectors, and all pixel values are rescaled to gray scale. We examine several different network structures, including architectures of 1 hiddel layer with 1024 units, 2 hidden layers with 800 neurons each, as well as 3 hidden layers with 1024 units each.

Table 1 compares the performance of our proposed models against other techniques. We use a learning rate of $0.01$ and decay it by $0.5$ after every $\{300, 400, 500\}$ epochs. We let all models run sufficiently long with $100K$ updates. For all models, we also explore different initialization for neuron retaining rates, including $\{0.8, 1.0\}$ for input

| Model | 1024 | $800 \times 2$ | $1024 \times 3$ |
|---|---|---|---|
| Multilayer Perceptron | 1.69 | 1.62 | 1.61 |
| + Dropout | 1.22 | 1.28 | 1.25 |
| + VARDROP | 1.20 | 1.16 | 1.07 |
| + SPARSEVARDROP | 1.34 | 1.30 | 1.27 |
| + Rademacher | 1.11 | 1.08 | 0.95 |

Table 1: Classification error on MNIST dataset.

layers, $\{0.5, 0.8, 1.0\}$ for hidden layers. In practice, we find initializing the retaining rates to $0.8$ for input layer and $0.5$ for hidden layer yields better performance for all models, except for SPARSE-VARDROP model, initializing retaining rate to $1.0$ for input layer seems to give better result.

Figure 3 illustrates the changes in retain rates for both input and hidden layers under Rademacher regularization with $1e^{-4}$ regularization weight. The network contains one hidden layer of 1024 *ReLU* units. The retain rates were initialized to $0.8$ for input layer and $0.5$ for hidden layer. The learning rate is $0.01$ and decayed by half after every 200 epochs. We observe the retain rates for all layers are diffused throughout training process, and finally converged towards a bimodal distribution for input layer and a unimodal distribution for hidden layer. We also notice that the retain rates for input

layer upon model convergence demonstrate interesting feature pattern of the dataset. For example, the pixels in surrounding margins yield smaller retain rates, and the center pixels often have larger retain rates. This is because the digits in MNIST dataset are often centered in the image, hence all the surrounding pixels are not predictive at all when classifying an instance. This demonstrates that our proposed method is able to dynamically determine if an input signal is informational or not, and subsequently gives higher retain rate if it is, otherwise reduce the retain rate over time.

## 4.2 CIFAR

CIFAR10 and CIFAR100 datasets are collections of $50K$ training and $10K$ testing RGB images from 10 and 100 different image categories. Every instance consists of $32 \times 32$ RGB pixels. We preprocess all images by subtracting the per-pixel mean computed over all training set, then with ZCA whitening as suggested in Srivastava et al. (2014). No data augmentation is used. The neural network architecture we evaluate on uses three convolutional layers, each of which followed by a max-pooling layer. The convolutional layers have 96, 128, and 256 filters respectively. Each convolutional layer has a $5 \times 5$ receptive field applied with a stride of 1 pixel, and each max-pooling layer pools from $3 \times 3$ pixel region with strides of 2 pixels. These convolutional layers are followed by two fully-connected layer having 2048 hidden units each.

Table 2 summarizes the performance of our proposed models against other baselines. We initialize dropout rates settings with $\{0.9, 1.0\}$ for input layers, $\{0.75, 1.0\}$ for convolutional layers and $\{0.5, 1.0\}$ for fully-connected layers. Similarly to the MNIST evaluation, we find setting the corresponding retaining probabilities for input layers, convolutional layers and fully-connected layers to 0.9, 0.75 and 0.5 respectively yields best performance under all models. We initialize the learning rate to 0.001 and decay it exponentially every $\{200, 300, 400\}$ epochs.

| Model | CIFAR10 | CIFAR100 |
|---|---|---|
| Convolutional neural network | 18.01 | 50.28 |
| + Dropout in fully-connected | 17.05 | 45.81 |
| + VARDROP | 16.85 | 45.47 |
| + SPARSEVARDROP | 17.87 | 45.74 |
| + Rademacher | 16.89 | 45.35 |
| + Dropout in all layers | 15.16 | 41.00 |
| + VARDROP | 15.03 | 39.15 |
| + SPARSEVARDROP | 15.87 | 42.67 |
| + Rademacher | 13.81 | 38.63 |

Table 2: Classification error on CIFAR datasets.

Figure 4 illustrates the changes in retain rates for both input and hidden layers under Rademacher regularization with 0.1 regularization weight. The network contains two convolution layers with 32 and 64 convolutional filters followed by one fully-connected layer with 1024 neurons. All hidden units use *ReLU* activation functions. The retain rates were initialized to 0.9 for input layer, 0.75 for convolutional layer and 0.5 for fully-connected layer. The learning rate is 0.001 and exponentially decayed by half after every 300 epochs. Similar to MNIST dataset, we observe the retain rates for all layers are diffused throughout training process, and finally converged towards a unimodal distribution. However, unlike MNIST dataset, we do not see similar pattern for retain rates of input layer. This is mainly due to the nature of dataset, such that CIFAR10 images spread over the entire range, hence all pixels are potentially informational to the classification process. This again demonstrates that the Rademacher regularizer is able to distinguish the informational pixels and retain them during training.

## 4.3 TEXT CLASSIFICATION

In addition, we also compare our proposed approach on text classification datasets—SUBJ and IMDB. SUBJ is a dataset containing $10K$ subjective and objective sentences (Pang & Lee, 2004) with nearly $14.5K$ vocabulary after stemming. All subjective comments come from movie reviews expressing writer's opinion, whereas objective sentences are from movie plots expressing purely facts. We randomly sample $20\%$ from the collections as test data, and use other $80\%$ for training and validation. IMDB is a collection of movie reviews from IMDB website, with $25K$ for training and another $25K$ for test (Maas et al., 2011), containing more than $50K$ vocabulary after stemming. It contains an even number of positive (i.e., with a review score of 7 or more out of a scale of 10) and negative (i.e., with a review score of 4 or less out of 10) reviews. The dataset has a good movie diversity coverage

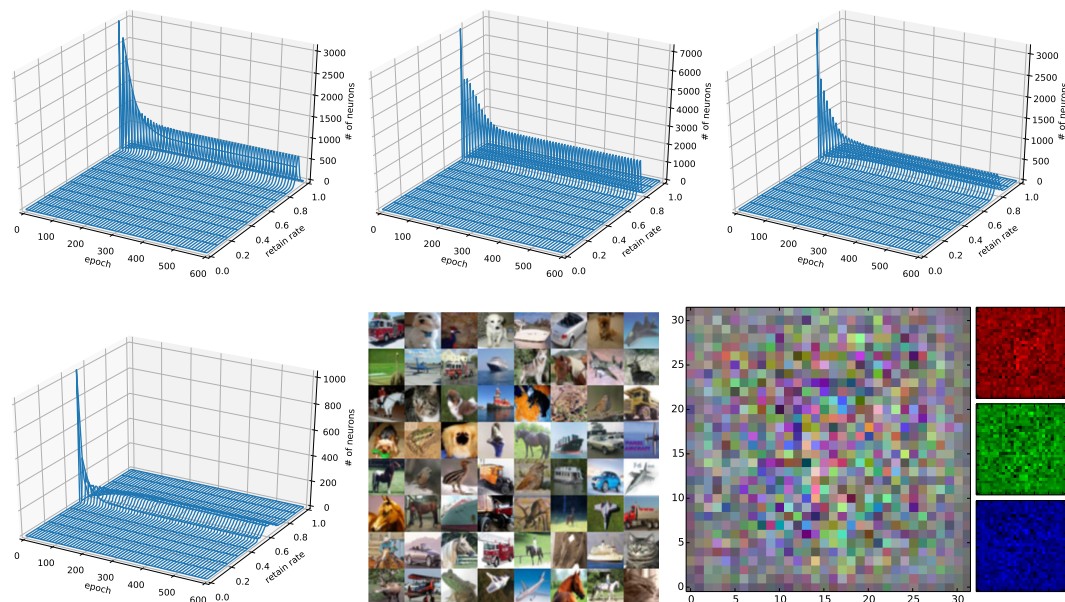

Figure 4: Changes in retain rates with Rademacher regularization on CIFAR10 dataset. Top-Left: changes in retain rate histograms for input layer ($32 \times 32 \times 3$ RGB pixels) through training. Top-Middle: changes in retain rate histograms for first convolutional layer ($32 \times 15 \times 15$ units) through training process. Top-Right: changes in retain rate histograms for second convolutional layer ($64 \times 7 \times 7$ units) through training process. Bottom-Left: changes in retain rate histograms for fully-connected layer (1024 *ReLU* units) through training process. Bottom-Middle: sample images from CIFAR10 dataset. Bottom-Right: retain rates for corresponding input pixels in both superposition and individual RGB channels upon model convergence. Unlike MNIST datasets, there is no clear pattern from the retain rates out of these channel pixels, since they are all informational towards prediction.

with less than 30 reviews per movie. For each sentence or document in these datasets, we normalize it into a vector of probability distribution over all vocabulary.

Table 3 summarizes the performance of our proposed models against other baselines. We initialize dropout rates settings with $\{0.8, 1.0\}$ for input layers and $\{0.5, 1.0\}$ for fully-connected layers. Similarly, by setting the corresponding retaining probabilities for input layers and fully-connected layers to $0.8$ and $0.5$ respectively, the model often yields the best performance. We use a constant learning rate of $0.001$, as well as an initialization learning rate of $0.01$ and decay it by half every $\{200, 300, 400\}$ epochs. We notice that overall the improvement of dropout is not as significant as MNIST or CIFAR datasets.

Figure 5 illustrates the changes in retain rates for both input and hidden layers under Rademacher regularization with $0.005$ regularization weight on IMDB dataset. The network contains one hidden layer of 1024 *ReLU* units. The retain rates were initialized to $0.8$ for input layer and $0.5$ for hidden layer. The learning rate is $0.01$ and decayed by half after every 200 epochs. Similar to other datasets, we observe the retain rates for all layers are diffused slightly upon model convergence, and in particularly the retain rates for input layer demonstrate interesting feature patterns of the data.

| Model | SUBJ | IMDB |
|---|---|---|
| Multi-layer Perceptron | 11.50 | 12.18 |
| + Dropout | 10.95 | 12.02 |
| + VARDROP | 10.45 | 11.82 |
| + SPARSEVARDROP | 10.35 | 11.97 |
| + Rademacher | 10.15 | 11.83 |

Table 3: Classification error on text dataset.

Recall that the task for IMDB dataset is to classify movie reviews into negative or positive labels. Generically speaking, adjectives are more expressive than nouns or verbs in this scenario, and our findings seems to be consistent with this intuition, i.e., yield high retain rates. List of the most indicative features include *"wonder(ful)"*, *"best"*, *"love"*, *"trash"*, *"great"*, *"classic"*, *"recommend"*, *"terribl(e)"*, *"perfect"*, *"uniqu(e)"*, *""fail*, *"amaz(ing)"*, *"fine"*, *"supris(e)"*, *"worst"*, *"silli(y)"*, *"flawless"*, *"wast(e)"*, *"dull"* and *"ridicul(ous)"*. As discussed above, nouns or verbs are

more often used to describe the movie plot, hence are less indicative, i.e., with smaller retain rates. Some of the word features with low retaining probability—hence, possibly less indicative—include *"year"*, *"young"*, *"possibl(e)"*, *"happen"*, *"dead"*, *"music"*, *"flick"*, *"shot"*, *"oscar"*, *"kill"*, *"spent"*, *"pretti(y)"*, *"say"*, *"review"*, *"support"*, *"anim(ation)"*, *"actual"*, *"call"*, *"cut"*, and *"role"*. One interesting observation is that we find the word *"oscar"* is also in the list of less informative features, which implies movie reviews and Academy Awards are not necessarily correlated.

In addition, we also include a list of popular and possibly unique named entities that are relevant to movie industry, including *"baldwin"*, *"niro"*, *"spacey"*, *"depp"*, *"downey"*, *"pitt"*, *"pacino"*, *"marilyn"*, *"hepburn"*, *"craig"*, *"dench"*, *"sammo"*, *"clooney"*, *"kidman"*, *"mccarthi"*, *"kermit"*, *"godzilla"*, *"nimoy"*, *"shawshank"*, *"yokai"*, *"emraan"*, *"kurosawa"*, *"spielberg"*, *"cameron"*, *"pacino"*, *"jackson"*, *"eastwood"*, *"allen"*, and *"verhoeven"*. We also notice interesting pattern on this list. For example, some actors (e.g., *"baldwin"* and *"kidman"*, etc.) and directors (e.g., *"kurosawa"* and *"eastwood"* etc.), yield high retain rates shortly after initial optimization. The retain rates of actors like *"downey"* or *"spacey"*, and directors like *"spielberg"* or *"cameron"* slightly increase or remain similar over time . The word *"pitt"*,[2] however, yields a declined retain rates throughout training, which suggests a less indicative feature for review classification. Note that higher retain rate means the corresponding features are more indicative in classifying IMDB reviews into positive or negative labels, i.e., no explicit association with the label itself.

## 5 CONCLUSION

Imposing regularizaiton for a better model generalization is not a new topic. However we tackle the problem for the dropout neural network regularization in a different way. The theoretical upper bound we proved on the Rademacher complexity facilitates us to directly incorporate the dropout rates into the objective function. In this way the dropout rate can be optimized by block coordinate descent procedure with one consistent objective. Our empirical evaluation demonstrates promising results and interesting patterns on adapted retain rates.

In the future, we would like to investigate the sparsity property of the learnt retain rates to encourage a sparse representation of the data and the neural network structure (Wen et al., 2016), similar to the sparse Bayesian models and relevance vector machine (Tipping, 2001). We would also like to explore the applications of deep network compression (Han et al., 2015a; Iandola et al., 2016; Ullrich et al., 2017; Molchanov et al., 2017; Louizos et al., 2017). In addition, one other possible research direction is to dynamically adjust the architecture of the deep neural networks (Srinivas & Babu, 2015; Han et al., 2015b; Guo et al., 2016), and hence reduce the model complexity via dropout rates.

### ACKNOWLEDGMENTS

We thank the anonymous reviewers and the program committee during ICLR review process for their insightful comments and valuable suggestions, in particular the third reviewer for pointing out an issue in proof on an earlier version of the manuscript. We would also like to show our gratitude to Li Wan for helpful discussions throughout this work. We are also immensely grateful to Jingwen Lu, Ciya Liao, Qifa Ke, Pinar Donmez and Shawn Chang for their support and guidance at Microsoft AI Research. Any findings, opinions and errors are solely on authors' responsibility, and should not tarnish the reputations of these esteemed persons, agencies and organizations.

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

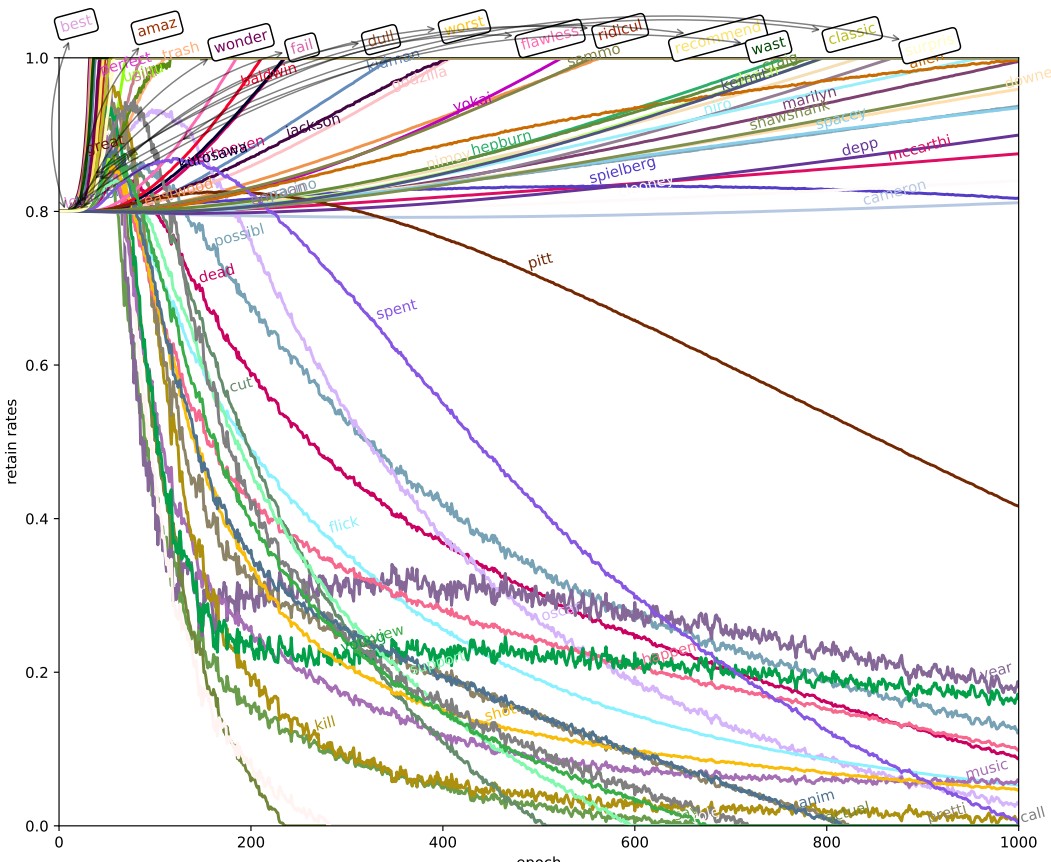

Figure 5: Changes in retain rates for different word features upon model convergence with Rademacher regularization on IMDB dataset. Features include words associated with the 20 largest and smallest retain rates, as well as a collection of movie related entities, e.g., actor, director, etc. From our model, words like *"love"*, *"great"*, *"terribl(e)"*, *"perfect"*, *"uniqu(e)"*, *"fine(st)"*, *"supris(e)"*, and *"silli(y)"* yield large retain rates and hence are indicative feature for predication (in the top half). On the other hand, words like *"say"*, *"pretti(y)"*, *"young"*, *"review"*, *"role"*, *"anim(ation)"* and *"actual"* have near zero retain rates upon model convergence, which are less informative. For named entities that are relevant to movie industry, we also observe some interesting pattern. Some actors (e.g., *"baldwin"* and *"kidman"*, etc.) and directors (e.g., *"kurosawa"* and *"eastwood"* etc.), yield high retain rates shortly after initialization. The retain rates of actors like *"downey"* or *"spacey"*, and directors like *"spielberg"* or *"cameron"* slightly increase or remain similar throughout optimization. Note that higher retain rate means the corresponding features are more indicative in classifying IMDB reviews into positive or negative labels, i.e., no explicit association with the label itself.

*of The 33rd International Conference on Machine Learning*, volume 48 of *Proceedings of Machine Learning Research*, pp. 1050–1059, New York, New York, USA, 20–22 Jun 2016. PMLR.

Wei Gao and Zhi-Hua Zhou. Dropout rademacher complexity of deep neural networks. *Science China Information Sciences*, 59(7):072104, Jun 2016. ISSN 1869-1919.

Ian J. Goodfellow, David Warde-farley, Mehdi Mirza, Aaron Courville, and Yoshua Bengio. Maxout networks. In *Proceedings of the International Conference of Machine Learning*, 2013.

Yiwen Guo, Anbang Yao, and Yurong Chen. Dynamic network surgery for efficient dnns. In *Advances In Neural Information Processing Systems*, pp. 1379–1387, 2016.

Song Han, Huizi Mao, and William J Dally. Deep compression: Compressing deep neural networks with pruning, trained quantization and huffman coding. *arXiv preprint arXiv:1510.00149*, 2015a.

Song Han, Jeff Pool, John Tran, and William Dally. Learning both weights and connections for efficient neural network. In *Advances in Neural Information Processing Systems*, pp. 1135–1143, 2015b.

Trevor Hastie, Robert Tibshirani, and Jerome Friedman. *The elements of statistical learning: data mining, inference and prediction*. Springer, 2 edition, 2009.

Forrest N Iandola, Song Han, Matthew W Moskewicz, Khalid Ashraf, William J Dally, and Kurt Keutzer. Squeezenet: Alexnet-level accuracy with 50x fewer parameters and< 0.5 mb model size. *arXiv preprint arXiv:1602.07360*, 2016.

Diederik P Kingma, Tim Salimans, and Max Welling. Variational dropout and the local reparameterization trick. In C. Cortes, N. D. Lawrence, D. D. Lee, M. Sugiyama, and R. Garnett (eds.), *Proceedings of Advances in Neural Information Processing Systems*, pp. 2575–2583. Curran Associates, Inc., 2015.

Alex Krizhevsky, Ilya Sutskever, and Geoffrey E Hinton. Imagenet classification with deep convolutional neural networks. In *Advances in neural information processing systems*, 2012.

Zhe Li, Boqing Gong, and Tianbao Yang. Improved dropout for shallow and deep learning. In D. D. Lee, M. Sugiyama, U. V. Luxburg, I. Guyon, and R. Garnett (eds.), *Advances in Neural Information Processing Systems 29*, pp. 2523–2531. Curran Associates, Inc., 2016.

Christos Louizos, Karen Ullrich, and Max Welling. Bayesian compression for deep learning. *arXiv preprint arXiv:1705.08665*, 2017.

Andrew L. Maas, Raymond E. Daly, Peter T. Pham, Dan Huang, Andrew Y. Ng, and Christopher Potts. Learning word vectors for sentiment analysis. In *Proceedings of the Association for Computational Linguistics*, HLT '11, pp. 142–150, Stroudsburg, PA, USA, 2011. Association for Computational Linguistics. ISBN 978-1-932432-87-9.

Shin-ichi Maeda. A bayesian encourages dropout. *arXiv preprint arXiv:1412.7003*, 2014.

David A. McAllester. A pac-bayesian tutorial with A dropout bound. *CoRR*, abs/1307.2118, 2013.

Dmitry Molchanov, Arsenii Ashukha, and Dmitry Vetrov. Variational dropout sparsifies deep neural networks. In Doina Precup and Yee Whye Teh (eds.), *Proceedings of the 34th International Conference on Machine Learning*, volume 70 of *Proceedings of Machine Learning Research*, pp. 2498–2507, International Convention Centre, Sydney, Australia, 06–11 Aug 2017. PMLR.

Vinod Nair and Geoffrey E. Hinton. Rectified linear units improve restricted boltzmann machines. In Johannes Fürnkranz and Thorsten Joachims (eds.), *Proceedings of the International Conference of Machine Learning*, pp. 807–814. Omnipress, 2010.

Bo Pang and Lillian Lee. A sentimental education: Sentiment analysis using subjectivity summarization based on minimum cuts. In *Proceedings of the Association for Computational Linguistics*, Proceedings of the Association for Computational Linguistics, Stroudsburg, PA, USA, 2004. Proceedings of the Association for Computational Linguistics.

Shai Shalev-Shwartz and Shai Ben-David. *Understanding Machine Learning: From Theory to Algorithms*. Cambridge University Press, New York, NY, USA, 2014. ISBN 1107057132, 9781107057135.

Suraj Srinivas and R Venkatesh Babu. Learning neural network architectures using backpropagation. *arXiv preprint arXiv:1511.05497*, 2015.

Suraj Srinivas and R Venkatesh Babu. Generalized dropout. *arXiv preprint arXiv:1611.06791*, 2016.

Nitish Srivastava, Geoffrey Hinton, Alex Krizhevsky, Ilya Sutskever, and Ruslan Salakhutdinov. Dropout: A simple way to prevent neural networks from overfitting. *The Journal of Machine Learning Research*, 15(1), 2014.

Ilya Sutskever, James Martens, George Dahl, and Geoffrey Hinton. On the importance of initialization and momentum in deep learning. In Sanjoy Dasgupta and David McAllester (eds.), *Proceedings of the International Conference of Machine Learning*, volume 28 of *Proceedings of Machine Learning Research*, pp. 1139–1147, Atlanta, Georgia, USA, 17–19 Jun 2013.

Michael E Tipping. Sparse bayesian learning and the relevance vector machine. *Journal of machine learning research*, 1(Jun):211–244, 2001.

Karen Ullrich, Edward Meeds, and Max Welling. Soft weight-sharing for neural network compression. *arXiv preprint arXiv:1702.04008*, 2017.

Stefan Wager, Sida Wang, and Percy S Liang. Dropout training as adaptive regularization. In *Advances in neural information processing systems*, 2013.

Li Wan, Matthew Zeiler, Sixin Zhang, Yann L Cun, and Rob Fergus. Regularization of neural networks using dropconnect. In *Proceedings of the International Conference of Machine Learning*, 2013.

Sida Wang and Christopher Manning. Fast dropout training. In *Proceedings of the 30th International Conference on Machine Learning*, 2013.

Wei Wen, Chunpeng Wu, Yandan Wang, Yiran Chen, and Hai Li. Learning structured sparsity in deep neural networks. In *Advances in Neural Information Processing Systems*, pp. 2074–2082, 2016.

Jingwei Zhuo, Jun Zhu, and Bo Zhang. Adaptive dropout rates for learning with corrupted features. In Qiang Yang and Michael Wooldridge (eds.), *International Joint Conference on Artificial Intelligence*. AAAI Press, 2015.

## 6 APPENDIX

### 6.1 PROOF OF THEOREM 3.1

*Proof.* In the analysis of Rademacher complexity, we treat the functions fed into the neurons of the $l^{\text{th}}$ layer as one function class $\mathfrak{F}^l = f^l(\mathbf{x}; \mathbf{w}^{:l})$. Here again we are using the notation $\mathbf{w}^{:l} = \{\mathbf{w}^1, \ldots, \mathbf{w}^l\}$, and $\mathbf{w} = \mathbf{w}^{:L}$. As a consequence $\forall j, f_j^l(\mathbf{x}; \mathbf{w}^l) \in \mathfrak{F}^{\mathfrak{l}}$.

Note here $f^L(x; W)$ used in section 3 is a vector, but $f^L(x; w)$ used in this subsection is a scalar. The connection between $f^L(x; W)$ and $f^L(x; w)$ is that each dimension of $f^L(x; W)$ is viewed as one instance coming from the same function classs $f^L(x; w)$. Similar ways of proof have been adopted in Wan et al. (2013).

To simplify our analysis, we follow Wan et al. (2013) and reformulate the cross-entropy loss on top of the softmax into a single logistic function

$$\text{loss}(f^L(\mathbf{x}; \mathbf{W}), \mathbf{y}) = - \sum_j y_j \log \frac{e^{f_j^L(\mathbf{x}; \mathbf{W})}}{\sum_j e^{f_j^L(\mathbf{x}; \mathbf{W})}}.$$

The function class fed into the neurons of the $l^{th}$ layer $f^l(x; w^{:l})$ admits a recursive expression

$$f^l(x; w^{:l}, r^{:(l-1)}) = \sum_k \phi(f_k^{l-1}(x; w^{:l-1}, r^{:l-2})) r_k^{l-1} w_k^l \tag{4}$$

$$f^l(x; w^{:l}; \theta^{:(l-1)}) = E_{r^{:(l-1)}} f^l(x; w^{:l}, r^{:(l-1)}) \tag{5}$$

Given the neural network function (1) and the logistic loss function $l$ is 1 Lipschitz, by Contraction lemma (a variant of the lemma 26.9 on page 381, Chapter 26 of (Shalev-Shwartz & Ben-David, 2014)), the empirical Rademacher complexity of the loss function is bounded by

$$R_{\mathbb{S}}(l \circ f^L) = \frac{1}{n} E_\sigma max_w \sum_i \sigma_i l(f^L(x_i; w), y_i)$$

$$\leq \frac{k}{n} E_\sigma max_w \sum_i \sigma_i f^L(x_i; w) = k R_{\mathbb{S}}(f^L) \tag{6}$$

Note the empirical Rademacher complexity of the function class of the $L^{th}$ layer, i.e., the last output layer, is

$$R_{\mathbb{S}}(f^L) = \frac{1}{n} E_{\{\sigma_i\}} \left[ \sup_w \sum_i \sigma_i f^L(x_i; w) \right] \tag{7}$$

To prove the bound in a recursive way, let's also define a variant of the Rademacher complexity with absolute value inside the supremum:

$$\tilde{R}_{\mathbb{S}}(f) = \frac{1}{n} E_{\{\sigma_i\}} \sup_w \left| \sum_i \sigma_i f(x_i; w) \right| \tag{8}$$

Note here $\tilde{R}_{\mathbb{S}}(f)$ is not exactly the same as the Rademacher complexity defined before in this paper. And we have

$$R_{\mathbb{S}}(f^L) \le \tilde{R}_{\mathbb{S}}(f^L) \tag{9}$$

Now we start the recursive proof. The empirical Rademacher complexity (with absolute value inside supremum) of the function class of the $l^{th}$ layer is

$$\tilde{R}_{\mathbb{S}}(f^l) = \frac{1}{n} E_{\{\sigma_i\}} \sup_w \left| \sum_i \sigma_i f^l(x_i; w) \right|$$
$$= \frac{1}{n} E_{\{\sigma_i\}} \sup_w \left| \sum_i \sigma_i E_{r:l-1} f^l(x_i; w, r) \right| \tag{10}$$

Let

$$\hat{R}_{\mathbb{S}}(f^l) = E_{r:l-1} \left[ \frac{1}{n} E_{\{\sigma_i\}} \sup_w \left| \sum_i \sigma_i f^l(x_i; w, r) \right| \right] \tag{11}$$

By the calculous of Rademacher complexity,

$$\tilde{R}_{\mathbb{S}}(f^L) = \tilde{R}_{\mathbb{S}}(E_{r:l-1} f^L(x; w, r)) = \tilde{R}_{\mathbb{S}}(\sum_r p(r) f^L(x; w, r)) \le \sum_r p(r) \tilde{R}_{\mathbb{S}} f^L(x; w, r) = \hat{R}_{\mathbb{S}}(f^L) \tag{12}$$

Now we have

$$\hat{R}_{\mathbb{S}}(f^l) = E_{r:l-1} \left[ \frac{1}{n} E_{\{\sigma_i\}} \sup_w \left| \sum_i \sigma_i f^l(x_i; w, r) \right| \right]$$
$$= E_{r:l-1} \left[ \frac{1}{n} E_{\{\sigma_i\}} \sup_w \left| \sum_i \sigma_i \sum_j w_j^l r_j^{l-1} \phi(f_j^{l-1}(x_i; w^{:l-1}, r^{:l-2})) \right| \right] \tag{13}$$

Let $g_j^{l-1}(x; w^{:l-1}, \theta^{:(l-2)}) = \phi\left(f_j^{l-1}(x_i; w^{:l-1}, r^{:l-2})\right)$, then

$$\hat{R}_{\mathbb{S}}(f^l) \le E_{r:l-1} \left[ \frac{1}{n} E_{\{\sigma_i\}} \sup_{w^{:l-1}} \sup_{w^l} \left| \sum_i \sigma_i \sum_j w_j^l r_j^{l-1} g_j^{l-1}(x_i; w^{:l-1}, r^{:(l-2)}) \right| \right] \tag{14}$$

According to the assumption, $p \geq 1$, $q = p/(p-1)$, and $\|w^l\|_p \leq B^l$, from equation (14), by Holder's inequality, we have $\forall l \in \{2, \ldots, L\}$

$$\hat{R}_{\mathbb{S}}(f^l) \leq E_{r^{:l-1}} \left\{ \frac{1}{n} E_{\{\sigma_i\}} \left[ \sup_{\|w^l\|_p \leq B^l} |w^l \odot r^{l-1}|_1 \sup_{w^{:l-1}} \sup_j | \sum_i \sigma_i g_j^{l-1}(x_i; w^{:l-1}, r^{:(l-2)})| \right] \right\}$$

$$\leq E_{r^{:l-1}} \left\{ \left( \sup_{\|w^l\|_p \leq B^l} \sum_j r_j^{l-1} |w_j^l| \right) \frac{1}{n} E_{\{\sigma_i\}} \sup_{w^{:l-1}} \sup_j \left| \sum_i \sigma_i g_j^{l-1}(x_i; w^{:l-1}, r^{:(l-2)}) \right| \right\}$$

$$\leq E_{r^{:l-1}} \left\{ \left( \sup_{\|w^l\|_p \leq B^l} \|r^{l-1}\|_q \|w^l\|_p \right) \tilde{R}_{\mathbb{S}}(g^{l-1}(x; w^{:l-1}, r^{:l-2})) \right\}$$

$$\leq B^l \|\theta^{l-1}\|_1^{1/q} E_{r^{:l-2}} \left[ \tilde{R}_{\mathbb{S}}(g^{l-1}(x; w^{:l-1}, r^{:l-2})) \right] \tag{15}$$

In the last inequality above we used Jensen's inequality since $q \geq 1$ as well as the fact that the dropout random variable $r_j$ is binary (so $r_j^q = r_j$).

Suppose the activation function $\phi(\cdot)$ used in the neural network is 1-Lipschitz, and $\phi(0) = 0$ (for example, the RELU function). Then by the Ledoux-Talagrand contraction lemma, $\forall l \in \{1, \ldots, L-1\}$, given $r$,

$$\tilde{R}_{\mathbb{S}}(g^{l-1}(x; w^{:l}, r^{:l-1})) \leq 2\tilde{R}_{\mathbb{S}}(f^{l-1}(x; w^{:l-1}, r^{:l-2})). \tag{16}$$

From equation (16) and (15) we have

$$\hat{R}_{\mathbb{S}}(f^l) \leq 2B^l \|\theta^{l-1}\|_1^{1/q} \hat{R}_{\mathbb{S}}(f^{l-1}) \tag{17}$$

For the first layer, i.e., the feature layer with dropout but without activation function, if $|w^1|_p \leq B^1$, then

$$\hat{R}_{\mathbb{S}}(f^1(x; w^1)) = E_{r^0} \left\{ \frac{1}{n} E_{\{\sigma_i\}} \left[ \sup_{\|w^1\|_p \leq B^1} \left| \sum_{i=1}^n \sigma_i < w^1, x_i \odot r^0 > \right| \right] \right\}$$

$$= E_{r^0} \left\{ \frac{1}{n} E_{\{\sigma_i\}} \left[ \sup_{\|w^1\|_p \leq B^1} \left| \sum_{i=1}^n \sigma_i \sum_j w_j^1 x_{ij} r_j^0 \right| \right] \right\}$$

$$= E_{r^0} \left\{ \frac{1}{n} E_{\{\sigma_i\}} \left[ \sup_{\|w^1\|_p \leq B^1} \left| \sum_j w_j^1 r_j^0 \sum_{i=1}^n \sigma_i x_{ij} \right| \right] \right\}$$

$$\leq E_{r^0} \left\{ \frac{1}{n} E_{\{\sigma_i\}} \left[ \sup_{\|w^1\|_p \leq B^1} \|w^1 \odot r^0\|_1 \| \sum_{i=1}^n \sigma_i x_i\|_\infty \right] \right\}$$

$$\leq \frac{B^1 \|\theta^0\|_1^{1/q}}{n} E_{\{\sigma_i\}} \left[ \| \sum_{i=1}^n \sigma_i x_i\|_\infty \right] \tag{18}$$

By Lemma 26.11 on page 383, Chapter 26.2 of Shalev-Shwartz & Ben-David (2014), we know the last term in (18) is bounded by

$$\frac{1}{n} E_{\{\sigma_i\}} \left[ \| \sum_{i=1}^n \sigma_i x_i\|_\infty \right] \leq \max_i \|x_i\|_\infty \sqrt{2\log(2d)/n}.$$

Thus we get

$$\hat{R}_{\mathbb{S}}(f^1(x; w^1)) \leq \max_i \|x_i\|_\infty B^1 \|\theta^0\|_1^{1/q} \sqrt{\frac{2\log(2d)}{n}} \tag{19}$$

Combining the inequalities (6), (9), (15), (16), and (19), we have

$$R_{\mathbb{S}}(l \circ f^L) \leq k2^L \sqrt{\frac{2\log(2d)}{n}} \max_i \|x_i\|_\infty \left( \Pi_{l=1}^L B^l \|\theta^{l-1}\|_1^{1/q} \right). \tag{20}$$

$\square$

## 6.2 GENERALIZATION BOUND ON THE DROPOUT NEURAL NETWORKS

Here we need to define truncated cross entropy loss function:

$$\tilde{l} \circ f^L = \tilde{l}(f^L, (\mathbf{x}, \mathbf{y})) = \min(\text{loss}(f^L(\mathbf{x}; \mathbf{W}, \boldsymbol{\theta}), \mathbf{y}), C_l) \tag{21}$$

where $C_l$ is a constant. Note with the truncation, the cross entropy loss is still 1-Lipschitz so the empirical Rademacher complexity bound still holds for the truncated loss $\tilde{l}(f^L(\mathbf{x}; \mathbf{W}, \boldsymbol{\theta}), \mathbf{y})$.

**Theorem 6.1.** *For the dropout neural network defined in section (3), if truncated cross entropy loss $\tilde{l}$ (21) is used, then $\forall \delta \geq 0$, with probability at least $1 - 2\delta$, $\forall \tilde{l} \circ f^L$ :*

$$\mathbb{E}_{(x,y)\sim\mathbb{D}}(\tilde{l}(f^L(x; \mathbf{W}, \boldsymbol{\theta}), y) \leq \frac{1}{n} \sum_i \tilde{l}(f^L(\mathbf{x_i}; \mathbf{W}, \boldsymbol{\theta}), \mathbf{y_i}) + R_{\mathbb{S}}(\tilde{l} \circ f^L) + 3C_l\sqrt{1/(2n\delta)} \tag{22}$$

Note here the empirical Rademacher complexity for the bounded loss function $R_{\mathbb{S}}(\tilde{l} \circ f^L)$ admits the same bound as the empirical Rademacher complexity for the unbounded cross entropy loss $R_{\mathbb{S}}(l \circ f^L)$.

## 6.3 TOWARDS AN UNIFIED VIEW OVER RADEMACHER REGULARIZATION*

In fact, adding a Rademacher related regularizer, though not investigated much, is not new at least for linear functions.

It is well known (Shalev-Shwartz & Ben-David, 2014) that the empirical Rademacher complexity of the linear class

$$H_2 = \{x \to \langle w, x \rangle : \|w\|_2 \leq B_2\}$$

is bounded by

$$R_{\mathbb{S}} \leq \max_i \|x_i\|_2 B_2 / \sqrt{n}.$$

Note the $l_2$ loss function is 2-Lipschtz. In this way, we may interpret the regularizer in the ridge regression related an upper bound for the empirical Rademacher complexity of the linear function class.

Similarly for the linear class

$$H_1 = \{x \to \langle w, x \rangle : \|w\|_1 \leq B_1\},$$

the empirical Rademacher complexity is bounded by

$$R_{\mathbb{S}} \leq \max_i \|x_i\|_\infty B_1 \sqrt{\frac{2\log(2d)}{n}}.$$

So the lasso problem can also be viewed as adding a Rademacher-related regularization to the empirical loss minimization objective.

## 6.4 SCALES OF REGULARIZATION*

In application we do not want the regularization term to vary too much when the neural network has different number of internal neurons. To overcome that we design some heuristics to add to the regularizer. Note here all the scales mentioned in this section are added in a heuristic fashion. It is purely empirical.

When $p = q = 2$, the regularizer is bounded by

$$R_{\mathbb{S}}(l \circ f^L) \leq k2^L \sqrt{\frac{\log(d)}{n}} \max_i \|x_i\|_\infty \left( \Pi_{l=1}^L \max_j \|\mathbf{W}_j^l\|_2 \|\boldsymbol{\theta}^{l-1}\|_1^{1/2} \right). \tag{23}$$

---

*Note that the content in Section 6.3 and 6.4 is based purely on heuristics, and derived on an ad hoc basis.

where $\mathbf{W}_j^l \in \mathbb{R}^{k^{l-1}}$ is the $j$-th column of $\mathbf{W}^l$. Suppose we use i.i.d. uniform random variable to initialize $\mathbf{W}^l$ such that $\forall l \; w_{ij}^l \sim U[-\sqrt{6/(k^l + k^{l+1})}, \sqrt{6/(k^l + k^{l+1})}]$. Considering the scales of the maximum among the 2-norms of $\mathbf{W}_j^l$, we use a scaled regularizer instead:

$$k2^L \sqrt{\frac{\log d}{n}} \max_i \|x_i\|_\infty \left( \Pi_{l=1}^L \frac{\max_j \|\mathbf{W}_j^l\|_2 \|\boldsymbol{\theta}^{l-1}\|_1^{1/2}}{k^l} \sqrt{\frac{k^l + k^{l-1}}{\log k^l}} \right).$$

Similarly, when $p = \infty$ and $q = 1$, the scaled regularizer we used is

$$k2^L \sqrt{\frac{\log d}{n}} \max_i \|x_i\|_\infty \left( \Pi_{l=1}^L \|\mathbf{W}^l\|_{max} \|\boldsymbol{\theta}^{l-1}\|_1 \frac{\sqrt{k^l + k^{l-1}}}{k^l} \right).$$

### 6.5 Stability of the Dropout Rates Convergence

In this sub-section we demonstrate the dropout rates convergence across multiple runs. We use a neural network with one hidden layer of 1024 *ReLU* units to illustrate, and feed it with MNIST dataset. We train the network for 10 different runs, with same configurations and empirical settings, except different initializations on the network weight coefficients. Figure 6 shows the histogram of hidden layer retain rates upon model convergence under different runs. We observe similar dropout behavior and distribution among multiple runs upon model convergence, i.e., the histograms of the retain rates do not diverge much across different runs in regards to different model weight initializations.

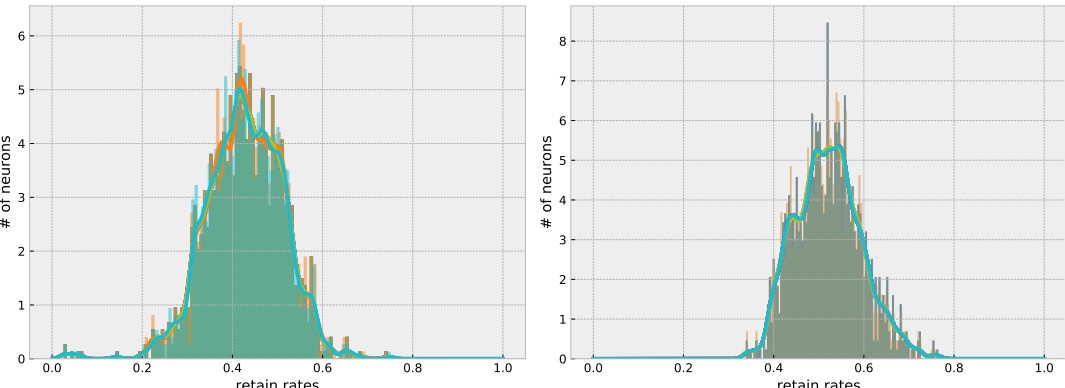

Figure 6: Histograms of the hidden layer dropout retain rates upon model convergence on 10 different runs under different Rademacher $p = \infty, q = 1$ regularization settings. The neural network contains one hidden layer of 1024 *ReLU* units, and is trained on MNIST dataset with different regularizer weight of $1e^{-3}$ (Left) and $1e^{-4}$ (Right). We train the neural network for 200 epochs, with a minibatch size of 100, initial learning rate of 0.01, and decay it by half every 40 epochs. The experiments are run with same configurations and experimental settings, except different initializations on the network weight coefficients.

