# OpenReview forum: "Adaptive Dropout with Rademacher Complexity Regularization"
_ICLR.cc/2018/Conference — Accept (Poster)_

### Official Review · AnonReviewer3 · 2017-11-15
**mathematical analysis seems not sound**

**Rating:** 6
**Confidence:** 3

**Review:**

This paper studies the adjustment of dropout rates which is a useful tool to prevent the overfitting of deep neural networks. The authors derive a generalization error bound in terms of dropout rates. Based on this, the authors propose a regularization framework to adaptively select dropout rates. Experimental results are also given to verify the theory.

Major comments:
(1) The Empirical Rademacher complexity is not defined. For completeness, it would be better to define it at least in the appendix.
(2) I can not follow the inequality (5). Especially, according to the main text, f^L is a vector-valued function . Therefore, it is not clear to me the meaning of \sum\sigma_if^L(x_i,w) in (5).
(3) I can also not see clearly the third equality in (9). Note that f^l is a vector-valued function. It is not clear to me how it is related to a summation over j there.
(4) There is a linear dependency on the number of classes in Theorem 3.1. Is it possible to further improve this dependency?

Minor comments:
(1) Section 4: 1e-3,1e-4,1e-5 is not consistent with 1e^{-3}, 1e^{-4},1e^{-5}
(2) Abstract: there should be a space before "Experiments".
(3) It would be better to give more details (e.g., page, section) in citing a book in the proof of Theorem 3.1

Summary:
The mathematical analysis in the present version is not rigorous. The authors should improve the mathematical analysis.

----------------------------
After Rebuttal:
Thank you for revising the paper. I think there are still some possible problems.
Let us consider eq (12) in the appendix on the contraction property of Rademacher complexity (RC).
(1) Since you consider a variant of RC with absolute value inside the supermum, to my best knowledge, the contraction property (12) should involve an additional factor of 2, see, e.g., Theorem 12 of "Rademacher and Gaussian Complexities: Risk Bounds and Structural Results" by Bartlett and Mendelson. Since you need to apply this contraction property L times, there should be a factor of 2^L in the error bound. This make the bound not appealing for neural networks with a moderate L.
(2) Second, the function g involves an expectation w.r.t. r before the activation function. I am not sure whether this existence of expectation w.r.t. r would make the contraction property applicable in this case.

---

> ### Author Response · Authors · 2017-12-15
> **some explanations on the questions raised by reviewer #3**
>
> Thanks very much for your review and comments.
>
> About your major comments
>
> (1):
> Thanks for your suggestion, we will include the definition of empirical Rademacher complexity in our revision.
>
> (2) and (3):
> As we stated in the first paragraph of our proof 6.1, we treat the functions fed into the neurons of the l-th layer as one class of functions. Therefore, f^L(x;W) is a vector as you correctly pointed out, but f^L(x;w) is a scalar. So each dimension of f^L(x;W) is viewed as one instance coming from the same function class f^L(x;w). Similar ways of proof have been adopted in Wan et al. (2013). We are sorry about the confusion. We will add more descriptions about it to make that clear in our revision.
>
> (4)
> It is a good question. The dependency on the number of classes comes from the contraction lemma. However, what we proved is only a weak bound on the Rademacher complexity. We are still working on further tightening the bound. For now, we are not sure if we can reduce the dependency on the number of classes to sub-linear. We hope this work will also open additional research directions and future extensions to the community. You are always welcome to add to it.
>
> About your minor comments:
>
> (1) (2) Thanks for the careful examination. We will fix the typos in the next version.
>
> (3) Thanks for the comments.
> Contraction lemma (Shalev-Shwartz & Ben-David, 2014) is a variant of the Lemma 26.9 located on page 381, Chapter 26.
> Lemma 26.11 in Shalev-Shwartz & Ben-David (2014) is located on page 383, Chapter 26.2.
> We will add the chapters and pages to the proof.

---

> ### Author Response · Authors · 2018-01-12
> **About the questions on the rebuttal raised by Reviewer #3:**
>
> Thanks very much for your careful examination. We do appreciate it.
> (1) If you look at the final Rademacher complexity bound we are proving, it has no absolute value inside the supremum. The contraction lemma is applied to the Rademacher complexity without absolute value. That is why equation (7) comes after the contraction. We understand this is confusing. We will make it clear in the next version.
> (2) As you mentioned, if we take expectation with respect to r, then f^L is not a function of r any more. Actually in our definition, the final prediction function f^L is a deterministic function (since we take the expectation w.r.t. r).

---

> > ### Comment · AnonReviewer3 · 2018-01-25
> > **response to the authors**
> >
> > Thanks for the response. In eq (12), the left-hand side is E \sup |E  \phi(f)|, while the right-hand side is E \sup |f|.
> > Here E denotes the expectation. Firstly, there is an absolute value operator here which requires a factor of 2 in the application of contraction property. Secondly, an expectation is inside the absolute value operator, so, as far as i can see, the standard contraction property can not be applied in this way .

---

> > > ### Author Response · Authors · 2018-01-26
> > > **The issues pointed out by reviewer 3 are valid. We fixed the issues in our updated draft.**
> > >
> > > Thanks very much for pointing out the issues. We are sorry in the last rebuttal we missed your point. Both issues you suggested are valid.
> > >
> > > For your first concern:
> > > Yes there is an absolute value operator inside, so this requires a factor of 2. Suppose we have L layers, there should be a factor of 2^L as you correctly pointed out.
> > > Fortunately, in our empirical evaluation L is fixed, so 2^L is a constant. In this way it won’t affect the experimental observation.
> > >
> > > For your second concern:
> > > You are right, the original procedure has issues due to the existence of E inside \sup. We fixed the issue in our latest revision. The final claim in the theorem stays the same (except there is an extra 2^L term as you suggested earlier). Please check our updated proof.
> > >
> > > We sincerely thank reviewer 3 for the great contribution and efforts.

---

> > > > ### Comment · AnonReviewer3 · 2018-01-26
> > > > **thanks for the update of the draft**
> > > >
> > > > Thank you for considering the comments in the revision. Here are some further comments:
> > > >
> > > > in eq (12), the last two equations are the same and one can be removed.
> > > > in eq (14), it seems that \|\theta\|_q should be \|\theta\|_1^{1/q}? Please check the deductions:
> > > > E\|r\|_q = E [ [\sum_i|r_i|^q]^{1/q} ] = E [ [\sum_i r_i ]^{1/q} ] \leq [\sum_i E r_i]^{1/q} = \|\theta\|_1^{1/q}

---

> > > > > ### Author Response · Authors · 2018-01-26
> > > > > **Thanks for your insightful comments.**
> > > > >
> > > > > You are right. After the update, the upper bound blows a little bit from \|\theta\|_q to \|\theta\|_1^{1/q}. The claim of the theorem now only holds when q = 1.  When q>1, we need to adjust the bound. We updated our draft to make sure the theory part is sound.
> > > > >
> > > > > Since we also have some experiments on q = 2 and q=\infty, this suggests part of our experiments needs to be modified and rerun. At this point we are considering withdraw and resubmit later.

---

> > > > > > ### Comment · AnonReviewer3 · 2018-01-27
> > > > > > **thanks for the update of the draft**
> > > > > >
> > > > > > Thanks for the revision addressing my concerns on the deduction. Also, feel regretful that the change of arguments leads to a different bound.

---

> > > > > > > ### Author Response · Authors · 2018-02-06
> > > > > > > **we updated the experiments**
> > > > > > >
> > > > > > > Your comments and suggestions are always welcome. We just updated the experiments with \|\theta\|_1^{1/q} so that they are consistent with the theorem. Please feel free to add more comments if you have further concerns. Thank you.

---

### Official Review · AnonReviewer1 · 2017-11-27
**This is an important piece of work that relates complexity of networks' learnability to dropout rates in backpropagation. This paper answers some critical questions about dropout learning.**

**Rating:** 7
**Confidence:** 5

**Review:**

An important contribution. The paper is well written. Some questions that needs to be better answered are listed here.
1. The theorem is difficult to decipher. Some remarks needs to be included explaining the terms on the right and what they mean with respect to learnability or complexity.
2. How does the regularization term in eq (2) relate to the existing (currently used) norm based regularizers in deep network learning? It may be straight forward but some small simulation/plots explaining this is important.
3. Apart from the accuracy results, the change in computational time for working with eq (2), rather than using existing state-of-the-art deep network optimization needs to be reported? How does this change vary with respect to dataset and network size (beyond the description of scaled regularization in section 4)?
4. Confidence intervals needs to be computed for the retain-rates (reported as a function of epoch). This is critical both to evaluate the stability of regularizers as well as whether the bound from theorem is strong.
5. Did the evaluations show some patterns on the retain rates across different layers? It seems from Figure 3,4 that retain rates in lower layers are more closer to 1 and they decrease to 0.5 as depth increases. Is this a general pattern?
6. It has been long known that dropout relates to non-negative weighted averaging of partially learned neural networks and dropout rate of 0.5 provides best dymanics. The evaluations say that clearly 0.5 for all units/layers us not correct. What does this mean in terms of network architecture? Is it that some layers are easy to average (nothing is learned there, so dropped networks have small variance), while some other layers are sensitive?
7. What are some simple guidelines for choosing the values of p and q? Again it appears p=q=2 is the best, but need confidence intervals here to say anything substantial.

---

> ### Author Response · Authors · 2017-12-15
> **some thoughts about the questions raised by reviewer #1**
>
> Thanks very much for your encouraging comments and helpful suggestions.
>
> 1. The upper bound suggests that layers affect the complexity in a multiplicative way. An extreme case as we described in the last paragraph of section 3.2 is, if the dropout retain rates for one layer are all zeros, then the empirical Rademacher complexity for the whole network is zero since the network is doing random guess for predictions. In this case the bound is tight. We will put more descriptions about the terms in our bound.
>
> 2. This is an interesting suggestion. Norm based regularizers currently used are imposed on the weights of each layer without considering the retain rates and the regularization is done on each layer independently. We suggest organizing them in a systematic way.
>
> 3. In terms of running time, the proposed framework takes one additional backpropagation compared to its standard deep network counterpart. In practice, we find the running time per epoch after introducing the regularizer is approximately 1.6 to 1.9 times that of the current standard deep network.
>
> 4: Thanks for the great suggestions.  There are some potential issues with drawing the confidence intervals for the retain rate of a particular neuron. For example, permuting the neurons does not change the network structure but it may lead to some identifiability issues. Instead to demo the stability of the algorithm we may add a plot showing the histograms of the theta with different initializations.
>
> 5. This is an excellent question! In fact, we are conducting additional evaluations to verify this pattern. We had some preliminary empirical observations that, as the layer goes higher, fewer neurons get high retain rates . This is somewhat consistent with the fact that people tend to set the number of neurons smaller for higher layers. We still need more experiments to tell if this is a general pattern.
>
> 6. This is another great question. It is also related to an on-going follow-up work we are currently investigating as stated in the conclusion and future work section of our paper. If we use the setting of p=\infty and q=1, the L1 norm regularizer may produce sparse retain rates. Subsequently, we could prune the corresponding neuron. Therefore we could use the algorithm as a way to determine the number of neurons used in hidden layers, i.e., we can use the regularizer to tune the network architecture. Similarly, if we use p=1 and q=\infty, then we can expect sparse coefficients on W due to the property of the L1 norm, in this way the regularizer can also be used to prune the internal neural connections.
>
> 7. Currently we do not have any theory for choosing p and q. As we stated above, one way is to choose p and q based on the sparsity desire. If we would like to impose sparsity on the number of neurons to fire,we may set q=1 to promote sparse retain rates. On the other hand, if we would like to impose sparsity on the number of internal connections, i.e., have a sparse coefficient matrix W, we may set p=1 instead.

---

### Official Review · AnonReviewer2 · 2017-11-28
**does not actual say why using Rademacher as a regularizer is theoretically justified, and why the loose bound is reasonable**

**Rating:** 6
**Confidence:** 3

**Review:**

==Main comments

The authors connect dropout parameters to a bound of the Rademacher complexity (Rad) of the network. While it is great to see deep learning techniques inspired by learning theory, I think the paper makes too many leaps and the Rad story is ultimately unconvincing.  Perhaps it is better to start with the resulting regularizer, and the interesting direct optimization of dropout parameters. In its current form, the following leaps problematic and were not addressed in the paper:

1) Why is is adding Rad as a regularizer reasonable? Rad is usually hard to compute, and most useful for bounding the generalization error. It would be interesting if it also turns out to be a good regularizer, but the authors do not say why nor cite anything. Like the VC dimension, Rad itself depends on the model class, and cannot be directly optimized. Even if you can somehow optimize over the model class, these quantities give very loose bounds, and do not equal to generalization error. For example, I feel even just adding the actual generalization error bound is more natural. Would it make sense to just add Rad to the objective in this way for a linear model?

2) Why is it reasonable to go from a regularizer based on RC to a loose bound of Rad? The actual resulting regularizer turns out to be a weight penalty but this seems to be a rather loose bound that might not have too much to do with Rad anymore. There should be some analysis on how loose this bound is, and if this looseness matter at all.

The empirical results themselves seem reasonable, but the results are not actually better than simpler methods in the corresponding tasks, the interpretation is less confident. Afterall, it seems that the proposed method had several parameters that were turned, where the analogous parameters are not present in the competing methods. And the per unit dropout rates are themselves additional parameters, but are they actually good use of parameters?

==Minor comments

The optimization is perhaps also not quite right, since this requires taking the gradient of the dropout parameter in the original objective. While the authors point out that one can use the mean, but that is more problematic for the gradient than for normal forward predictions. The gradient used for regular learning is not based on the mean prediction, but rather the samples.

tiny columns surrounding figures are ugly and hard to read

---

> ### Author Response · Authors · 2017-12-15
> **some comments about the questions raised by reviewer#2**
>
> Thanks very much for your valuable comments and helpful suggestions.
>
> Q: Why adding Rad as a regularizer reasonable? Why is it reasonable to go from a regularizer based on RC to a loose bound of Rad?
> A: These are great questions. We agree we do not have a rigorous way to prove adding an approximate upper bound to the objective can lead to any theoretical guarantee as you correctly pointed out.  The theorem of the upper bound in the paper is rigorous but why adding the upper bound to the objective can help is heuristic and empirical.
>
> On the other hand, adding an approximate term that is related to the upper bound of the Rademacher complexity is not something new. For example, the squared L2 norm regularizer used in the ridge regression, though there are explanations such as Bayesian priors, can be interpreted as a term related to the upper bound of the Rademacher complexity of linear classes . People are already using it. Similarly, the L1 regularizer used in LASSO can also be interpreted as a term related to the Rademacher complexity bound. We have put a section in the Appendix (Section 6.3) to somewhat justify it in a heuristic way.
>
> Q: The actual resulting regularizer turns out to be… rather loose bound…
> A: We agree that the bound proved in the paper could be a bit loose. Still in some extreme cases it is tight. For example, as we indicated in the paragraph before Section 3.3, if the retain rates in one layer are all zeros, the model always makes random guess for prediction. In this case the empirical Rademacher complexity is zero and our bound is tight. In general, even if the bound is loose, it still gives some justification on the norms used in today’s neural network regularizations. Additionally, it leads to a systematic way of weighting the norms as well as the retain rates.
>
> Minor comments:
> Q: While the authors point out that one can use the mean, but that is more problematic for the gradient than for normal forward predictions. After all, the gradient used for regular learning is not based on the mean prediction, but rather the samples.
> A: This is an excellent question. As we stated in Section 3.3, “this is an approximation to the true f^L(x;W, θ)”. Using the mean is purely an approximation used for the sake of optimization efficiency. By design we should use the samples. However empirically we found that optimizing based on the mean (instead of the actual sampling) still leads to a decrease of the objective. We will add additional figures to better illustrate the point in our next revision.
>
> Q: tiny columns surrounding figures are ugly and hard to read
> A: Thanks for the suggestion. We will fix it in our revision.
>
> Q: dropout rate is perhaps more common than retain rate
> A: We use the retain rate instead just to make the upper bound look less messy.

---

> > ### Comment · AnonReviewer2 · 2018-01-24
> > **explain/motivate convincingly why adding Rademacher complexity itself is reasonable**
> >
> > Not motivating why adding Rademacher itself is in my opinion the biggest weakness in the paper.
> > While section 6.3 is a good step (though I think it belong to the main paper given the story), it still seems inadequate. For one, the hypothesis is now: bounds of Rad makes good regularizers, instead of Rademacher makes good regularizers. Secondly, for L1, Rad is the infinity norm, which is not the one you wanted.
> > Lastly, Rad only depends on the hypothesis class and not on how much data your have and properties of the data, which are clearly important for picking regularizers in practice (or their strength through cross validation), which suggests it might not be justifiable.
> >
> > I still think the paper is borderline and weak reject unless this issue can be addressed convincingly.

---

> > > ### Author Response · Authors · 2018-01-24
> > > **more explanations on the concerns by Reviewer #2**
> > >
> > > Thanks very much for the comments. We would like to clarify some misunderstandings:
> > >
> > > Q: for L1, Rad is the infinity norm, which is not the one you wanted.
> > > A: section 6.3, for L1, Rad contains the 1 norm (B_1). The L_infty norm is on the samples, just like the max norm shown in our bound. Note here B1 instead of L_infty is the bound on the model parameters.
> > >
> > > Q: Rad only depends on the hypothesis class and not on how much data your have and properties of the data
> > > A: This is wrong. Not only does Rademacher complexity depend on the hypothesis class, but it also depends on the sample distribution because it is taking the expectation of the empirical Rademacher complexity over all samples of size n.
> > > On the other hand, note what we proved is the upper bound for the EMPIRICAL Rademacher complexity rather than the Rademacher complexity itself. That is why it has the dependency on the sample size.
> > > By measure concentration the EMPIRICAL Rademacher complexity is used to bound the Rademacher complexity.
> > >
> > > Q: the hypothesis is now: bounds of Rad makes good regularizers, instead of Rademacher makes good regularizers.
> > > A:  As stated in section 6.3, the regularizer used in ridge regression as well as Lasso can be interpreted as terms related to the upper bound of the empirical Rademacher complexity. Rademacher itself may make a good regularizer, but it is simply too complex to optimize and evaluate. That’s why we used the upper bound instead.
> > > Similar ways of approximation have also been used in the numeric optimization community, where if the objective is too hard to optimize, one may choose to optimize its convex envelop instead.

---

> > > > ### Comment · AnonReviewer2 · 2018-01-24
> > > > **correcting misunderstanding**
> > > >
> > > > (it looks like my previous reply did not appear, so trying again)
> > > > Thanks for correcting my mistakes on both the L1 norm and the data dependence. It seems plausible in that adding Rad is justifiable. I think 6.3 is important in the chain of reasoning. Right now, it is suggestive but does not sufficiently justify the approach. It would be valuable to verify that Rad give a reasonable dependence both in terms of n and d as a regularizer. Secondly, it still suggests that bounds of Rad seems to be common regularizers, rather than that Rad itself is good.
> > > >
> > > > However, given that it is intuitive to optimize a generalization bound, and my previous misunderstanding, I changed my rating to weak accept.

---

### Author Response · Authors · 2018-01-05
**We have posted a revision of the draft**

Update list:

1. added a subsection 6.5 to the appendix to empirically demonstrate the relations between the stochastic objective and the deterministic approximation. (minor comments from Reviewer#2)
2. added a subsection 6.6 to the appendix to empirically show the stability of the dropout rate convergence. (suggestion 4 from Reviewer #1)
3. added one paragraph of descriptions of the terms used in our bound following the theorem 3.1 as suggested by Reviewer #1 (Q1).
4. added two cases when our bounds are tight. (the last paragraph in section 3.1) This responds to the second concern raised by Review #2.
5. added the definition of the empirical Rademacher complexity to section 3.1 as suggested by Reviwer #3.
6. added a paragraph about the different notations used for vectors and scalars in subsection 6.1. (second paragraph of the proof) This is to respond the questions 2 and 3 raised by Reviewer #3.
7. fixed all the typos pointed out by Reviewer #3. (comment 1 and 2 from Reviewer #3)
8. added references of pages and chapters suggested by Reviewer #3.
9. fixed the tiny column issues mentioned by Review #2
10. for the first concern raised by Review #2, we suggest reading section 6.3 in our appendix.

---

### Comment · Area_Chair · 2018-01-24
**PAC-Bayes prior work**

Are the authors aware of the work by David McAllister using PAC-Bayes bounds to analyze dropout?  Last I saw, it was not mentioned in the paper.  IT seems like important related work.  Could the authors, very quickly (!), comment as to the relationship and explain what, if any, changes they would make to address this gap in related work?

---

> ### Author Response · Authors · 2018-01-24
> **we will include a reference to David's work on the PAC-Bayes bound**
>
> Thanks so much for pointing out this fantastic work. Much appreciated. We apologize that we were not aware of the work by David McAllister before. We believe the work is definitely related to ours. We will add a reference in our next revision.
>
> The work by David provides a bound on the expected loss from a nice but different point of view.  Some differences from our work:
> 1. The PAC-Bayesian bound assumes a distribution over the hypothesis class. The Rademacher bound we proved does not make that assumption.
> 2. The bound David proved assumes one universal dropout rate so \alpha in their paper is a scalar. To tune the retain rate for each individual neuron, the retain probabilities we used in our bound are vectors. That is, we assume different neurons may have different dropout rates.

---

> > ### Comment · Area_Chair · 2018-01-25
> > **further thoughts**
> >
> > Re: 1.   The PAC-Bayes theorems hold for all "priors".  Each "prior" gives you a different bound. So it is not right to say that David "assumes" a prior. There is no assumption. In the application of PAC-Bayes bounds, priors are often chosen for convenience to yield tractable KL divergence terms. This is the case in David's bounds.
> >
> > Re: 2.  I suspect the extension of David's bound to different dropout rates is straightforward.

---

> > > ### Author Response · Authors · 2018-01-26
> > > **Thanks for the comments and suggestions.**
> > >
> > > We added David McAllister’s paper in our latest revision.
> > >
> > > 1. The PAC-Bayes theorem holds for all “priors”. The Rademacher bound holds for all "hypothesis" in the class. We do not assume there is any probability measure over the hypothesis class. But we agree adding priors often gives us convenience in the proof.
> > >
> > > 2. Thanks for the great suggestion. Extension of David’s bound to different dropout rates is definitely worth trying.
> > >
> > > ############################################################################
> > >
> > > The issue pointed out by reviewer 3 is valid. Though we fixed the issue, it leads to a slight change of the bound when q > 1. As a result, some of the experiments with the setting q > 1 need to be modified and rerun.
> > >
> > > At this point we are considering withdraw and resubmit.
> > >
> > > Thanks very much for your time and effort.
> > >
> > > ############################################################################
> > > We updated the experiments and now they are consistent with the updated theorem. Thanks for your patience and understanding.

---

### Decision · Program_Chairs · 2018-01-29
**ICLR 2018 Conference Acceptance Decision**

**Decision:**

Accept (Poster)

**Comment:**

The reviewers agreed that the work addresses an important problem. There was disagreement as to the correctness of the arguments in the paper: one of these reviewers was eventually convinced. The other pointed out another two issue in their final post, but it seems that 1. the first is easily adopted and does not affect the correctness of the experiments and 2. the second was fixed in the second revision. Ideally these would be rechecked by the third reviewer, but ultimately the correctness of the work is the authors' responsibility.

Some related work (by McAllister) was pointed out late in the process. I encourage the authors to take this related work seriously in any revisions. It deserves more than two sentences.